# An Investigation of Structure-Activity Relationships of Azolylacryloyl Derivatives Yielded Potent and Long-Acting Hemoglobin Modulators for Reversing Erythrocyte Sickling

**DOI:** 10.3390/biom10111508

**Published:** 2020-11-02

**Authors:** Abdelsattar M. Omar, Osheiza Abdulmalik, Mohini S. Ghatge, Yosra A. Muhammad, Steven D. Paredes, Moustafa E. El-Araby, Martin K. Safo

**Affiliations:** 1Department of Pharmaceutical Chemistry, Faculty of Pharmacy, King Abdulaziz University, Alsulaymanyah, Jeddah 21589, Saudi Arabia; ymuhammad@kau.edu.sa (Y.A.M.); madaoud@kau.edu.sa (M.E.E.-A.); 2Department of Pharmaceutical Chemistry, Faculty of Pharmacy, Al-Azhar University, Cairo 11884, Egypt; 3Division of Hematology, The Children’s Hospital of Philadelphia, Philadelphia, PA 19104, USA; abdulmalik@email.chop.edu; 4Department of Medicinal Chemistry, School of Pharmacy and Institute for Structural Biology, Drug Discovery and Development, Virginia Commonwealth University, Richmond, VA 23219, USA; msghatge@vcu.edu (M.S.G.); sdpar97@gmail.com (S.D.P.)

**Keywords:** hemoglobin, sickle cell disease, Michael addition, antisickling, oxygen equilibrium

## Abstract

Aromatic aldehydes that bind to sickle hemoglobin (HbS) to increase the protein oxygen affinity and/or directly inhibit HbS polymer formation to prevent the pathological hypoxia-induced HbS polymerization and the subsequent erythrocyte sickling have for several years been studied for the treatment of sickle cell disease (SCD). With the exception of Voxelotor, which was recently approved by the U.S. Food and Drug Administration (FDA) to treat the disease, several other promising antisickling aromatic aldehydes have not fared well in the clinic because of metabolic instability of the aldehyde moiety, which is critical for the pharmacologic activity of these compounds. Over the years, our group has rationally developed analogs of aromatic aldehydes that incorporate a stable Michael addition reactive center that we hypothesized would form covalent interactions with Hb to increase the protein affinity for oxygen and prevent erythrocyte sickling. Although, these compounds have proven to be metabolically stable, unfortunately they showed weak to no antisickling activity. In this study, through additional targeted modifications of our lead Michael addition compounds, we have discovered other novel antisickling agents. These compounds, designated MMA, bind to the α-globin and/or β-globin to increase Hb affinity for oxygen and concomitantly inhibit erythrocyte sickling with significantly enhanced and sustained pharmacologic activities in vitro.

## 1. Introduction

Sickle cell disease (SCD) is a genetic hematologic disorder affecting over 100,000 people in the U.S. and several millions worldwide [1,2]. The disease occurs as a result of a replacement of a polar residue βGlu6 with a hydrophobic βVal6 residue, changing normal Hb (HbA) into sickle Hb (HbS). When deoxygenated or under hypoxic conditions, the pathogenic βVal6 residue on one HbS tetramer makes a hydrophobic interaction with an adjacent HbS tetramer, polymerizing into long and rigid fibers, which is further stabilized by several intermolecular interactions between the HbS tetramers, causing sickling of red blood cells (RBCs) [3,4,5,6,7,8]. The rigid RBCs lead to several interrelated secondary pathophysiological events, including but not limited to, impaired blood flow, hemolysis, painful vaso-occlusion (VOC), adhesion of RBCs to tissue endothelium, oxidative stress, decreased vascular nitric oxide (NO) bioavailability, inflammation, and eventually chronic endothelial and organ damage that ultimately leads to poor quality of life and decreased life expectancy [1,2,9,10,11].

Currently, there are four approved therapies for SCD. The first is hydroxyurea (HU), which works by inducing fetal Hb (HbF) production to inhibit HbS polymerization [12]. A reported lack of response in about 30% of patients, partly due to poor compliance, tend to limit its use [13,14]. In 2017, L-glutamine (Endari), which reduces oxidative stress was approved to treat SCD [15]. However, it has limited efficacy [16,17]. In 2019, two other drugs, Crizanlizumab (Adakveo)[18] and Voxelotor [19] (GBT-440 or Oxbryta) were approved. The former is a monoclonal antibody that targets P-selectin to reduce the frequency of painful vaso-occlusive crises [18]. Voxelotor, an aromatic aldehyde binds to Hb and prevents HbS polymerization by increasing Hb oxygen (O_2_) affinity [19,20,21,22].

Hb functions in equilibrium between the unliganded or deoxygenated (Deoxy) tense (T) state, which exhibits low affinity for ligand, and an ensemble of oxygenated relaxed states (collectively referred to as R state) that exhibit high affinity for oxygen [23,24,25]. A plot of partial pressure and Hb oxygen saturation gives a sigmoidal oxygen equilibrium curve (OEC). The degree of shift in the OEC is reported as a decrease (left-shift) or increase (right-shift) in P_50_ (oxygen tension at 50% Hb O_2_ saturation). Since only deoxygenated T-state HbS polymerizes, allosteric modulation of Hb with compounds that destabilize the T-state and/or stabilize the R-state to increase Hb O_2_ affinity could prevent the hypoxia-induced HbS polymerization and sickling of RBC [23,24,25,26,27,28,29,30,31,32,33].

Several compounds, including aromatic aldehydes, e.g., Voxelotor, 5-HMF, vanillin, and their derivatives [20,21,22,23,24,25,26,27,28,29,30,31,32,33], thiols [34,35], and Michael addition compounds [36,37], are known to covalently bind to Hb and pharmacologically increase its oxygen affinity and/or directly destabilize the polymer to prevent RBC sickling. Aromatic aldehydes form Schiff base covalent interactions with the N-terminal αVal1nitrogen at the α-cleft of Hb, while thiols and Michael addition compounds, in most part react covalently with βCys93 of Hb. Aromatic aldehydes have been the most studied class of compounds, with Voxelotor approved for the treatment of SCD [19]. Nonetheless, unlike Michael addition compounds, most aromatic aldehydes undergo significant and rapid metabolic oxidation of the aldehyde moiety [38,39,40], resulting in poor pharmacokinetic (PK) properties, hampering their development into viable therapeutic agents.

To overcome the limitation of aromatic aldehydes susceptibility to oxidative metabolism, our group has over the years embarked on the development of Michael addition compounds [36,37]. Although, these compounds as expected demonstrated metabolic stability, they showed weaker antisickling potencies compared to the aromatic aldehydes [36,37]. In this report, we have further refined the structures of these earlier Michael addition compounds, resulting in some of the compounds exhibiting a significant improvement in their pharmacologic properties. 

## 2. Materials and Methods

### 2.1. Study Approvals 

At Virginia Commonwealth University (VCU), normal whole blood was collected from adult donors (>18 years) after informed consent, in accordance with regulations of the IRB for Protection of Human Subjects (IRB #HM1) by the Institutional Review Board at VCU. At the Children’s Hospital of Philadelphia (CHOP), leftover blood samples from individuals with homozygous SS who had not been recently transfused, were obtained and utilized based on an approved IRB protocol (IRB# 11-008151) by the Institutional Review Board at CHOP, with informed consent. All experimental protocols and methods were performed in accordance with institutional (VCU and CHOP) regulations. 

### 2.2. Chemistry

Melting point was performed using OptiMelt Automated Melting Point System Digital Image; Processing Technology SRS, Stanford Research Systems (Sunnyvale, CA, USA). NMR spectroscopy was recorded on Bruker AVANCE III 400 (Bruker, Fällanden, Switzerland). LCMS spectroscopy was performed on Agilent Technologies 1260 Infinity LC/MSD system with DAD\ELSD Alltech 3300 and Agilent; LC\MSD G6120B mass-spectrometer (Santa Clara, CA, USA). High-resolution mass spectroscopy (HRMS) was performed by separation and mass spectrometric detection techniques and were performed with an Infinity 1260 UHPLC system (Agilent Technologies, Waldbronn, Germany) coupled to a 6224 Accurate Mass TOF LC/MS system (Agilent Technologies, Singapore). Chromatographic separation was achieved using Agilent Zorbax C18 column (Agilent Technologies, Santa Clara, CA, USA) (100mm × 2.1mm, 1.9 μm particle size). Mobile phase A consisted of 0.1% formic acid in water and mobile phase B consisted of 0.1% formic in acetonitrile. Injection volume of the samples solutions was 1μL. Separation was performed at a constant flow rate of 0.4ml/min at 40 °C. A linear gradient started at 5% mobile phase B and ramped to 95% in 6.5 min, flushed 1.5 min at 95% B. The column was re-equilibrated for 2 min with 5% mobile phase A. Positive ion mass spectra of the column eluate were recorded in the range of *m*/*z* 100–1500 at a measuring frequency of 9500 transients/s and a detection frequency of 4GHz. The Agilent ion source was operated using the following conditions: pressure of nebulizing gas (N_2_) was 40 psi; temperature and flow rate of drying gas (N_2_) was 320 °C and 6 L/min, respectively; The capillary voltage was set to 4 kV, the fragmentor potential to 120V and the skimmer potential to 75 V. Solvents and chemicals were HPLC grade. Water was purified by Millipore Water Purification System. All the chemicals, reagents and solvents had been purified and/or dried in according to well-known literature methods; vendors’ names: UORSY and Enamine (Kiev, Ukraine).

#### 2.2.1. Synthesis of the MMA-100 Series of Compounds 

##### Preparation of (E)-3-(1H-imidazol-2-yl)-1-(5-methoxypyridin-2-yl)prop-2-en-1-one (MMA-101)

To a solution of 5-methoxypicolinic acid (**1a**) (0.68 g, 3.58 mmol) in DCM (20 mL) was added 1,1’-carbonyldiimidazole (0.58 g, 3.58 mol) at 30 °C. The reaction mixture was stirred at this temperature for 3 h, whereupon N,O-dimethylhydroxylamine hydrochloride (0.349 g, 3.58 mmol) was added, and the solution stirred for 2 hr. The reaction mixture was poured into water (20 mL), and the organic layer separated, washed with brine (30 mL), dried over anhydrous sodium sulfate, and filtered. The filtrate was concentrated under reduced pressure to give *N*,5-dimethoxy-*N*-methylpicolinamide (**2a**) (0.632 g, 90% yield). 

Methylmagnesium bromide (3.2 M solution in 2-MethylTHF, 1.5 mL) was added via syringe to a solution of *N*,5-dimethoxy-*N*-methylpicolinamide (**2a**) (632 mg, 3.22 mmol) in THF (12 mL) under argon atmosphere at 0 °C. The resulting mixture was warmed to RT and left to stir for 10h. The resulting mixture was quenched with aq NH_4_Cl (8mL) at 0 °C and extracted with EtOAc (2 × 15 mL). Organic layers were combined, washed with brine (10 mL), dried over Na_2_SO_4_ and evaporated to give 1-(5-methoxypyridin-2-yl)ethan-1-one (**3a**) (0.432 g, 87% yield) for use without further purification.

The solution of NaOH (0.336 g, 8.4 mmol) in EtOH was added to the solution of 1-(5-methoxypyridin-2-yl)ethan-1-one (**3a**) (0.423 g, 2.8 mmol) and 1*H*-imidazole-2-carbaldehyde (0.268 g, 2.8 mmol) in EtOH (15 mL) under argon at rt. The reaction mixture was stirred for 24h. After this time 10% aqueous HCl was added at 0 °C to reaction mixture (for establish pH = 4.5) and then concentrated under reduced pressure. The resulting residue was dissolved in aq NaHCO_3_ and filtered to give MMA-101 (0.45 g, 70% yield). Melting point: 190–192 °C.^1^H-NMR (400 MHz, DMSO-*d*_6_) δ 8.44 (d, *J* = 2.9 Hz, 1H), 8.31 (d, *J* = 16.1 Hz, 1H), 8.11 (d, *J* = 8.7 Hz, 1H), 7.63 – 7.52 (m, 2H), 7.28 (s, 2H), 3.94 (s, 3H).^13^C-NMR (126 MHz, Chloroform-*d*) δ 187.68, 158.84, 146.86, 144.20, 137.54, 131.78, 124.72, 121.31, 120.90, 56.54. LC-MS (ESI), RT = 0.775 min, *m*/*z* 230.20 [M + H]^+^; HRMS (ESI), RT = 3.583 min, *m*/*z* 230.0934 [M + H]^+^, formula C_12_H_11_N_3_O_2_.

##### Preparation of (E)-3-(1H-imidazol-2-yl)-1-(5-methoxypyridin-3-yl)prop-2-en-1-one (MMA-102)

The compound was synthesized from (**1b**) following the procedure in MMA-101 synthesis, and gave (0.193g, 65% yield) MMA-102 with the following characterization profile. Melting point: 185–192 °C. ^1^H-NMR (400 MHz, DMSO-*d*_6_) δ 12.82 (s, 1H), 8.92 (d, *J* = 2.5 Hz, 1H), 8.28 (dd, *J* = 8.7, 2.5 Hz, 1H), 7.84 (d, *J* = 15.6 Hz, 1H), 7.50 (d, *J* = 15.6 Hz, 1H), 7.32 (s, 2H), 6.99 (d, *J* = 8.7 Hz, 1H), 3.96 (s, 3H);^13^C-NMR (126 MHz, DMSO-*d*_6_) δ 186.93, 166.65, 149.51, 143.87, 139.28, 132.16, 127.84, 121.47, 111.49, 54.44. LC-MS (ESI), RT = 0.806 min, *m*/*z* 230.0 [M + H]^+^; HRMS (ESI), RT = 0.55 min, *m*/*z* 230.0927 [M + H]^+^, formula C_12_H_11_N_3_O_2_.

##### Preparation of (E)-N-(4-(3-(1H-imidazol-2-yl)acryloyl)phenyl)methanesulfonamide (MMA-103)

NaOH (412 mg, 10.3 mmol) in aqueous solution (6 mL) was added drop-wise to a solution of N-(4-acetylphenyl)methanesulfonamide (**3c**) (730 mg, 3.43 mmol) and 1*H*-imidazole-2-carbaldehyde (330 mg, 3.43 mmol) in ethanol (8 mL) while stirring. The resulting reaction mixture was stirred at 60 **°**C for 7h. The ethanol was evaporated, followed by addition of water, and the mixture was neutralized by citric acid to pH ~7. The product was extracted with DCM, washed two times with water, and dried over Na_2_SO_4_. After evaporation of DCM, the product was purified by HPLC to give MMA-103 (150 mg, 15% yield). Melting point: 216–218 °C ^1^H-NMR (400 MHz, DMSO-*d*_6_) δ: 12.82 (s, 1H), 10.41 (s, 1H), 8.04 (d, *J* = 8.6 Hz, 2H), 7.85 (d, *J* = 15.6 Hz, 1H), 7.48 (d, *J* = 15.6 Hz, 1H), 7.35 (d, *J* = 8.6 Hz, 2H), 7.31 (s, 2H), 3.14 (s, 3H).^13^C-NMR (126 MHz, DMSO-*d*_6_) δ: 187.67, 143.96, 143.58, 132.63, 131.94, 130.51, 130.51 121.57, 118.18, 40.5. LC-MS (ESI), RT = 0.735 min, *m*/*z* 292.0 [M + H]^+^.

##### Preparation of (E)-3-(1H-imidazol-2-yl)-1-(4-(trifluoromethoxy)phenyl)prop-2-en-1-one (MMA-104)

(720mg, 3.54 mmol) of the ketone (**3d**) was reacted at room temperature for 7h following the same procedure used in MMA-103, and gave (150 mg, 15% yield) product MMA-104 with the following characterization profile. Melting point: 150–156.5 °C.^1^H-NMR (400 MHz, DMSO-*d*_6_) δ 12.89 (s, 1H), 8.17 (d, *J* = 8.4 Hz, 2H), 7.85 (d, *J* = 15.7 Hz, 1H), 7.58 (d, *J* = 8.3 Hz, 2H), 7.51 (d, *J* = 15.6 Hz, 1H), 7.33 (s, 2H).^13^C-NMR (126 MHz, DMSO-*d*_6_) δ 188.05, 151.97, 143.79, 136.81, 132.90 (2C), 131.13, 131.12, 131.0 (2C), 130.9, 121.34(2C). LC-MS (ESI), RT = 0.938 min, *m*/*z* 283.0 [M + H]^+^; HRMS (ESI), RT = 2.946 min, *m*/*z* 283.0696 [M + H]^+^, formula C_13_H_9_F_3_N_2_O_2_.

##### Preparation of (E)-5-(3-(1H-imidazol-2-yl)acryloyl)-2-methoxybenzonitrile (MMA-105)

1-(4-methoxyphenyl)ethan-1-one(**6**)(13.01 g, 0.086 mol, 1eq.) was added to solution of octane bis(tetrafluoroborate) (F-TEDA) (30 g, 0.086 mol, 1 eq.) and I_2_ (11 g, 0.043 mol, 0.5eq.) in 100 mL CH_3_CN, and the mixture stirred at room temperature for 16 h. The mixture was quenched with aqueous NaOH-5%, the organic layer separated, and the aqueous layer was extracted with EtOAc (3 × 30 mL). The organic layers were combined, washed with water (2 × 20 mL) and brine (2 × 15 mL), dried over Na_2_SO_4_, and concentrated in vacuum to give 8.53 g of 1-(3-iodo-4-methoxyphenyl)ethan-1-one (**7**) (yield = 61.3%). CuCN(4 g, 0.0465 mol, 1.5eq.) was added to solution of (**7**) (8.5 g, 0.0309 mol, 1 eq.) in 10 mL DMF, and the mixture stirred at 130 °C for 16 h. The reaction mixture was quenched with aqueous HCl-10%, the organic layer separated, and the aqueous layer was extracted with EtOAc (3 × 20 mL). The organic layers were combined, washed with water (2 × 10 mL) and brine (10 mL), dried over Na_2_SO_4_, and concentrated in vacuum to give (3.5 g, 65% yield) of 5-acetyl-2-methoxybenzonitrile (**3e**).

Subsequently, (**3e**) was reacted with the aldehyde (1 eq) and NaOH (1.3 eq.) following the procedure used in MMA-101. Water was poured to the reaction mixture, extracted with DCM (3 × 10 mL). The organic layers were combined, washed with water (3 × 10 mL), dried over Na_2_SO_4_, and concentrated in vacuum to give(yield = 61.3%) of the crude product. Purification by HPLC (15–25% water-ACN, flow: 40 mL/min (loading pump 5 mL/min ACN)) yielded (100 mg, 2%) of MMA-105, which had the following characterization profile. Melting point: 177–178.9 °C. ^1^H-NMR (400 MHz, DMSO-*d*_6_) δ 12.89 (s, 1H), 8.17 (d, *J* = 8.4 Hz, 2H), 7.85 (d, *J* = 15.7 Hz, 1H), 7.45 (d, *J* = 8.3 Hz, 1H), 7.30 (d, *J* = 15.6 Hz, 1H), 7.33 (s, 2H), 3.96 (s, 3H); LC-MS (ESI), RT = 0.840 min, *m*/*z* 254.2 [M + H]^+^; HRMS (ESI), RT = 0.745 min, *m*/*z* 254.0931 [M + H]^+^, formula C_14_H_11_N_3_O_2_.

##### Preparation of (E)-3-(1H-imidazol-2-yl)-1-(6-((tetrahydrofuran-3-yl)oxy)pyridin-3-yl)prop-2-en-1-one (MMA-106)

Compound1-(6-methoxypyridin-3-yl)ethan-1-one (**3b**) (0.432 g, 2.86 mmol) was dissolved in hydrogen bromide solution 33 wt.% in acetic acid and was refluxed for 48 hr. The reaction mixture was cooled and product filtered to give1-(6-hydroxypyridin-3-yl)ethan-1-one (**4b**) (0.305 g, 78% yield). 

To a solution of 1-(6-hydroxypyridin-3-yl)ethan-1-one (**4b**) (0.305 g, 2.23 mmol) in dry DMF (10 mL) was added Cs_2_CO_3_ (1.44 g, 4.46 mmol) and the reaction mixture stirred at RT for 30 min. Following, tetrahydrofuran-3-yl methanesulfonate (0.37 g, 2.23 mmol) was added at RT, and the reaction mixture was heated at 60 °C overnight. The resulting mixture concentrated under reduced pressure. The resulting residue taken up in water (30 mL) and extracted with EtOAc (3 × 15 mL), dried over anhydrous sodium sulfate, filtered, and concentrated under reduced pressure. The crude was purified by HPLC to give 1-(6-((tetrahydrofuran-3-yl)oxy)pyridin-3-yl)ethan-1-one (**5b**) (0.184 g, 41% yield). 

(**5b**) was reacted with the aldehyde and NaOH following the procedure for MMA-101, to give the product MMA-106 (0.126 g, 50% yield) after HPLC purification, which had the following characterization profile. Melting point 20 °C; ^1^H-NMR (400 MHz, DMSO-*d*_6_) δ 9.11 (t, *J* = 2.8 Hz, 1H), 8.79 (d, *J* = 15.0 Hz, 1H), 8.37 (dd, *J* = 8.7, 2.5 Hz, 1H), 7.78 (s, 2H), 7.57 (dd, *J* = 15.6, 3.5 Hz, 1H), 7.01 (d, *J* = 8.5 Hz, 1H), 5.71 – 5.57 (m, 1H), 4.02 – 3.70 (m, 4H), 2.36 – 2.16 (m, 1H), 2.10 – 1.96 (m, 1H); ^13^C-NMR (151 MHz, DMSO-*d*_6_) δ 185.98, 166.06, 150.31, 141.51, 139.63, 128.78, 127.21, 124.76, 122.57, 112.36, 77.32, 72.97, 66.82, 32.96. LC-MS (ESI), RT = 1.788 min, *m*/*z* 286.2 [M + H]^+^

##### Preparation of (E)-3-(1H-imidazol-2-yl)-1-(5-((tetrahydrofuran-3-yl)oxy)pyridin-2-yl)prop-2-en-1-one (MMA-107)

Compound MMA-107 was synthesized from (**3a**) following the procedure used for MMA-106, and gave (0.189 g, 50% yield) product with the following characterization profile. Melting point: 149–150 °C. ^1^H-NMR (400 MHz, DMSO-*d*_6_) δ 12.93 (s, 1H), 8.44 (d, *J* = 2.8 Hz, 1H), 8.31 (d, *J* = 16.1 Hz, 1H), 8.10 (d, *J* = 8.8 Hz, 1H), 7.60 (dd, *J* = 8.8, 3.0 Hz, 1H), 7.56 (d, *J* = 16.2 Hz, 1H), 7.29 (s, 2H), 5.27 (t, *J* = 5.3 Hz, 1H), 4.10 – 3.71 (m, 4H), 2.37–2.20 (m, 1H), 2.10–1.93 (m, 1H); ^13^C-NMR (151 MHz, DMSO-*d*_6_) δ 187.60, 156.87, 146.80, 144.03, 138.33, 131.62, 124.65, 122.51, 120.93, 72.55, 66.88, 32.73; LC-MS (ESI), RT = 0.757 min, *m*/*z* 286.2 [M + H]^+^; HRMS (ESI), RT = 1.494 min, *m*/*z* 286.1185 [M + H]^+^, formula C_15_H_15_N_3_O_3_.

#### 2.2.2. Synthesis of the MMA-200 Series of Compounds 

##### Preparation of 1-(4-methoxyphenyl)prop-2-en-1-one (MMA-202)

To a suspension of powdered aluminum chloride (0.723 g, 5.42 mmol) in dry DCM (40 mL) at 0 °C was added slowly 3-chloropropionyl chloride(0.455 mL, 4.74 mmol) and anisole (**8a**) (0.491 mL, 4.52 mmol) under cooling with water. The orange solution was stirred at room temperature for 3 h and left standing for about 12h. The dark orange solution was poured onto ice (20 g), the organic layer was separated, and the aqueous layer extracted two times with CH_2_Cl_2_. The collected organic phases were washed with water (2 × 20 mL), several times with aqueous NaOH (2%, 50 mL) until the organic phase did not turn back to yellow, and then twice with water (10–15 mL). The organic phase was dried over MgSO_4_ and the solvent removed under reduced pressure. Purification by gradient column chromatography [SiO_2_, CH_2_Cl_2_/petroleum ether (40/60), 20:1 to CH_2_Cl_2_] gave 0.923 g (97% yield) of 3-chloro-1-(4-methoxyphenyl)propan-1-one (**9a**) as a colorless solid. 

TEA (0.83 mL) was added to a solution of compound 3-chloro-1-(4-methoxyphenyl)propan-1-one (0.923 g, 4.64 mmol) **(9a)** in 20 mL CH_3_CN, stirred and refluxed for 1 hr. Then the reaction mixture was cooled, concentrated under vacuum, followed by addition of 50 mL H_2_O and extraction with DCM (3 × 10 mL). The organic layers were combined, washed with water (2 × 5 mL), dried over Na_2_SO_4_, filtrated and the solvent removed under reduced pressure at RT. Purification by gradient column chromatography [SiO_2_, CH_2_Cl_2_/petroleum ether (40/60), 20:1 to CH_2_Cl_2_] gave 0.562 g of MMA-202 (78% yield). Melting point 20 °C; ^1^H-NMR (400 MHz, Chloroform-*d*) δ 7.93 (d, *J* = 8.8 Hz, 2H), 7.14 (dd, *J* = 17.0, 10.5 Hz, 1H), 6.93 (d, *J* = 8.8 Hz, 2H), 6.39 (dt, *J* = 17.1, 1.5 Hz, 1H), 5.84 (dt, *J* = 10.5, 1.5 Hz, 1H), 3.84 (s, 3H); ^13^C-NMR (126 MHz, Chloroform-*d*) δ 189.21, 163.55, 132.14, 131.02, 130.21, 129.22, 113.85, 55.48; LC-MS (ESI), RT = 1.162 min, *m*/*z* 163.00 [M +H]^+^; HRMS (ESI), RT = 5.487 min, *m*/*z* 163.0752 [M + H]^+^, formula C_10_H_10_O_2_.

##### Preparation of 1-(4-methoxyphenyl)prop-2-yn-1-one (MMA-204)

To a solution of 4-methoxybenzoic acid (**11**) (200 mg, 1.31 mmol) dissolved in dry DCM (5 mL), 1,1’-carbonyldiimidazole (212.42 mg, 1.31 mmol) was added proportionally. Reaction mixture was proceeded by ultrasound (US) for 20 min. Next, N,O-dimethylhydroxylamine hydrochloride (193 mg, 1.97 mmol) and TEA (199.5 mg, 1.97 mmol) were added, and the reaction mixture was proceeded by US for 5h. The mixture was washed with water and purified by column chromatography (Eluent EtOAc\Hexane) to give N,4-dimethoxy-N-methylbenzamide (**12**) (189 mg, 73.8% yield).

Trimethylsilylacetylene (89 mg, 0.902 mmol) was dissolved in THF (5 mL) and cooled to 0 °C. To this solution was added n-BuLi as a 2.5%solution in hexane (0.4 mL, 0.902 mmol), dropwise. After 20 min, Weinreb amide (**12**) (160 mg, 0.820mmol) was added dropwise via syringe, and the reaction was allowed to slowly warm to 25 °C over 2 h. The reaction was diluted with Et_2_O, poured into saturated aqueous citric acid solution, and the aqueous phase was extracted twice by EtOAc. Organic phases were dried with Na_2_SO_4_, filtered, and solvent was evaporated. The residue was purified by flash chromatography (Eluent EtOAc\Hexane) to give 1-(4-methoxyphenyl)-3-(trimethylsilyl)prop-2-yn-1-one (**13**) (134mg, 70.3% yield).

1-(4-methoxyphenyl)-3-(trimethylsilyl)prop-2-yn-1-one (**13**) (250 mg, 1.08 mmol) was dissolved in THF (5 mL) and cooled to 0 °C, and potassium hydrofluoride (51 mg, 0.646 mmol) in 2 mL of water was then added. Reaction mixture was stirred for 5h, and the reaction was allowed to slowly warm to 25 °C over 2h. The reaction was diluted with CH_2_Cl_2_, the mixture was poured into saturated aqueous hydrochloric acid solution, and the aqueous phase was extracted twice by CH_2_Cl_2_. Organic phases were dried with Na_2_SO_4_, decanted, and solvent was evaporated. The residue was purified by flash chromatography (Eluent EtOAc\Hexane) to give MMA-204 (114 mg, 66% yield). Melting point 86 °C; ^1^H-NMR (400 MHz, Chloroform-*d*) δ 8.10 (d, *J* = 8.8 Hz, 2H), 6.93 (d, *J* = 8.8 Hz, 2H), 3.86 (s, 3H), 3.36 (s, 1H).^13^C-NMR (126 MHz, Chloroform-*d*) δ 175.94, 164.79, 132.15, 131.16, 129.60, 113.96, 80.41, 80.04, 55.63; HRMS (ESI), RT = 5.542 min, *m/z* 161.0597 [M + H]^+^, formula C_10_H_8_O_2_.

##### Preparation of 1-(4-((tetrahydro-2H-pyran-2-yl)oxy)phenyl)prop-2-en-1-one (MMA-205)

Phenol (**8b**) (3.65 g, 0.039 mol, 1 eq.) was dissolved in CH_2_Cl_2_ (20 mL) and cooled to 0 °C. AlCl_3_ (5.5 g, 0.0403 mol, 1.03 eq.) was added in small portions, keeping internal temperature below 5 °C. The reaction mixture was stirred for 10 minutes, followed by dropwise addition of a solution of 3-chloropropanoyl chloride (5 g, 0.0397 mol, 1.01 eq.) in CH_2_Cl_2_ (10 mL), and with stirring overnight at 0 °C. The reaction mixture was quenched with H_2_O (30 mL), the layers separated, and the water phase was extracted with CH_2_Cl_2_ (2 × 20 mL). The combined organic layers was washed with H_2_O (2 × 10 mL) and brine (2 × 10 mL), dried over Na_2_SO_4_ and evaporated. The obtained crude product was purified by column chromatography to give 3-chloro-1-(4-hydroxyphenyl)propan-1-one as orange oil **(9b**) (0.5 g, 7% yield).

To a precooled (0 °C) solution of (**9b**) (0.5 g, 2.7 mmol) in dihydropyran (5 mL), catalytical amount of TsOH*H_2_O was added and the mixture stirred overnight. Evaporation of the solvent gave the crude product of 3-chloro-1-(4-((tetrahydro-2*H*-pyran-2-yl)oxy)phenyl)propan-1-one (**10**) (1 g, 0.0017mol, about 45% yield), which was used in the next step without purification.

To the solution of **(10)** (crude 1 g, ~1.7mmol) in CH_2_Cl_2_ (10 mL) at 0 °C, a solution of TEA (1 g, 10mmol, 4–5eq.) in CH_2_Cl_2_ (10 mL) was added in a dropwise, followed by stirring overnight. Evaporation of the solvent and purification of the crude residue resulted in MMA-205 (120 mg, 0.5 mmol, 25% yield); ^1^H-NMR (500 MHz, Chloroform-*d*) δ 7.95 (d, *J* = 8.7 Hz, 2H), 7.17 (dd, *J* = 17.1, 10.6 Hz, 1H), 7.12 (d, *J* = 8.7 Hz, 1H), 6.43 (d, *J* = 17.0 Hz, 1H), 5.88 (d, *J* = 10.5 Hz, 1H), 5.53 (s, 1H), 3.91 – 3.80 (m, 1H), 3.66 – 3.59 (m, 1H), 2.07 – 1.95 (m, 1H), 1.90 (dd, *J* = 7.1, 3.5 Hz, 2H), 1.80 – 1.47 (m, 3H); ^13^C-NMR (126 MHz, Chloroform-*d*) δ 189.4 (s), 161.1 (s), 132.2 (s), 130.9 (s, J = 7.1 Hz), 129.3 (s), 116.1 (s), 96.1 (s), 62.1 (s), 30.1 (s), 25.1 (s), 18.5 (s); LC-MS (ESI), RT = 3.061 min, *m*/*z* 149.2 [M -C_5_H_9_O +H]^+^; HRMS (ESI), RT = 1.085 min, *m*/*z* 149.06174 [M -C_5_H_9_O +H]^+^, formula C_9_H_8_O_2_.

### 2.3. In Vitro Time-Dependent Hb Oxygen Equilibrium Studies Using Normal Whole Blood

To determine the metabolic stability of the compounds in whole blood, the MMA compounds were used to conduct time-dependent studies on Hb oxygen equilibrium using normal whole blood as previously described [30]. Briefly, blood samples (hematocrit 30%) in the presence of 2 mM concentration of compounds were incubated at 37 °C for 24 h with shaking (at 140 rpm). At 1.5, 3, 6, 8, 12, and/or 24h time intervals, aliquots of this mixture were removed and further incubated in TM8000 Thin film tonometer (Meon Medical Solutions) to equilibrate at oxygen tensions 6, 20, and 40 mmHg for approximately 10 minutes at 37 °C. The samples were then aspirated into an ABL 800 Automated Blood Gas Analyzer (Radiometer) to determine the pH, partial pressure of CO2 (pCO_2_), partial pressure of oxygen (pO_2_), and Hb oxygen saturation values (SO_2_). The measured values of pO_2_ (mmHg) and SO_2_ were then subjected to a non-linear regression analysis using the program Scientist (Micromath, Salt Lake City, UT) to estimate P_50_ as previously reported [30]. The observed P_50_ shifts values in %P_50_ shifts were plotted as function of time [hrs]. 

### 2.4. In Vitro Hemoglobin Modification, Oxygen Equilibrium and Antisickling Studies Using Human homozygous Sickle cell (SS) Blood 

The MMA series of compounds (MMA-101, MMA-102, MMA-103, MMA-104, MMA-105, MMA-106, MMA-107, MMA-202, MMA-204, and MMA-205) and the positive control vanillin were studied for their abilities to inhibit hypoxia-induced RBC sickling (RBC morphology study) and increase Hb oxygen as previously published [28,29,30,31,32,33]. Briefly, homozygous SS blood (hematocrit: 20%) suspensions were incubated under air in the absence or presence of 2 and 5 mM concentration of test compounds at 37 °C for 1h. Following, the suspensions were incubated under hypoxic condition (2.5% oxygen) at 37 °C for 2h. Aliquot samples were fixed with 2% glutaraldehyde solution without exposure to air, and then subjected to microscopic morphological analysis. The residual samples were washed in phosphate-buffered saline, and hemolyzed in hypotonic lysis buffer for the Hb oxygen equilibrium experiment. For the study, approximately 100 μL aliquot samples from the lysate were added to 4 mL of 0.1M potassium phosphate buffer, pH 7.0, in a cuvette and subjected to hemoximetry analysis using Hemox™ Analyzer (TCS Scientific Corp.) to assess P_50_ shifts and Hill coefficient values (n_50_). 

### 2.5. Reverse-Phase HPLC Studies to Determine which Hb Subunit Interacts with the Compounds 

To determine whether the MMA compounds bind to the α- and/or β-globin chains to effect the Hb oxygen binding property, hemolysates from the above antisickling studies with MMA-202, MMA-204, and MMA-205 (2 mM) were subjected to reversed-phase (RP) HPLC on a Hitachi D-7000 HSM Series, using a Jupiter 5 μm C-4 50 × 4.6 mm column, (Phenomenex®, Torrence, CA) and a gradient from 20% to 60% acetonitrile in 0.3% trifluoroacetic acid in 15 min, with UV detection at 215 nm as previously published [28].

### 2.6. Reactivity of MMA Compounds toward Csyteine Residues of Hb

The accessible sulfhydryl groups in Hb, and their reactivity with the MMA compounds, MMA-102, MMA-202 and MMA-204, were determined by following the exchange reaction of the thiols in Hb and DTNB (Ellman’s reagent) at 412 nm (*ε* = 14,150 M^−1^cm^−1^) as previously reported by us [36]. Briefly, an aqueous solution of Hb (50 μM in PBS) was mixed with MMA-102, MMA-202, MMA-204, or the positive control Ethacrynic acid (ECA) at 2 mM final concentration in a final volume of 500 μL. ECA is known to form covalent interaction with βCys93 of Hb [41]. The mixture was incubated at room temperature for 3 h, with shaking at 100 rpm, and then transferred to a microfiltration centrifugal tube (MWCO 10 kDa) and centrifuged at 7000 rpm for 30 min at 4 °C to separate Hb from excess reagents. Hb was washed with PBS and centrifuged again to a final volume of 100 μL. Following, 25 μL of each Hb solution was added to 0.1 M potassium phosphate buffer (475 μL) at pH 8.0 and incubated at 25 °C for 1 h (non-DTNB control samples). Another set of 25 μL Hb solution was added to phosphate buffer (465 μL) and 10 μL of DTNB (10 mM in buffer) and incubated at 25 °C for 1h. Before centrifuging the non-DTNB control samples, the absorbance of each sample was noted at 576 nm to determine the concentration of Hb in each sample. Both sets of tubes were centrifuged using different centrifugal filters (7000 rpm, 20 min, 4 °C) to collect the yellow filtrate (2-nitro-5-thiobenzoate), which was quantified by measuring absorbance at 412 nm.

## 3. Results and Discussion

### 3.1. Rationale for Design of MMA Compounds

Aromatic aldehydes, the most studied antisickling agents, form Schiff base through a reversible chemical reaction with the N-terminal αVal1 amine of the α-subunit to stabilize the R-state HbS and increase the protein affinity for oxygen [20,21,22,23,24,25,26,27,28,29,30,31,32,33]. Nonetheless, their susceptibility to oxidative metabolism by aldehyde dehydrogenase (ALDH), particularly in the liver and RBCs [38,39,40] leading to poor PK properties, have greatly hampered their development [30,33,42]. The chronic nature of SCD, as well as the large amount of Hb that needs to be modified to reach therapeutic dose, clearly requires long acting antisickling agent. To overcome the aldehyde moiety limitation, we rationally developed a new class of antisickling agents (imidazolylacryloyl derivatives; Figure 1) that incorporated a stable Michael addition reactive center that we hypothesized would form covalent interaction with the Hb β-subunit residue βCys93 to increase Hb oxygen affinity [36,37]. Although, these compounds were metabolically stable, they showed weak or no antisickling activity. Interestingly, crystallographic study of a representative compounds, KAUS-15 and KAUS-12, showed that these compounds instead of binding to βCys93, because of their carboxylate moiety, were being directed to the central water cavity of deoxygenated Hb to make covalent interaction with the N-terminal αVal1 amines of the two α-subunits [36]. Thus, unlike aromatic aldehyde that bind to liganded Hb and stabilize the R-state Hb to increase the protein affinity for oxygen, binding of the KAUS compounds to deoxygenated Hb ultimately led to no significant effect on Hb affinity for oxygen, and in some instances even decreased Hb affinity for oxygen. Accordingly, we reported a second generation of KAUS compounds by removing the acidic group (azolylacryloyl derivatives; Figure 1), expecting the novel compounds to direct and bind to βCys93. Expectedly, these compounds, as exemplified by KAUS-33 and KAUS-38, bind covalently to βCys93 [37].

KAUS-33 and KAUS-38 showed promising antisickling activities with long durations of action; however, they have limited solubility, and their potency was significantly less than aromatic aldehydes [37]. This potentially would require larger doses in order to achieve sufficient therapeutic effects. Since KAUS-33 and KAUS-38 were considered viable leads, we decided to refine and optimize these azolylacryloyl molecules to increase their biological activity while still maintaining the Michael addition reactive center for their metabolic stability (Figure 2). 

In the first modification (Figure 2), compounds MMA-101, MMA-102, MMA-106, and MMA-107 (imidazyl Michael addition compounds) have the benzene ring replaced with a pyridine ring to increase solubility and RBC distribution, as well as reactivity of the Michael addition moiety. Similar observations have been reported for aromatic aldehydes, where substitution of benzene with pyridine considerably increased the compounds reactivity and RBC distribution [20]. MMA-106 and MMA-107 are pyridyl derivatives of KAUS-33, where the acid-labile 2-tetrahydropyranyl moiety has been replaced with the 3-tetrahydrfuranyl, a smaller, polar but more stable group. The compounds MMA-103, MMA-104 and MMA-105, also imidazyl Michael addition compounds contain electron-withdrawing groups that are expected to decrease the electron density on the benzene ring (c.f. KAUS-33 and KAUS-38) aiming to investigate this electronic factor on the binding potency. In addition, the sulfonamide analogue MMA-103 is also able to form several hydrogen bonding as both acceptor and donor within the active site.

In another modification exercise to increase potency, we made a radical change to the scaffold with the aim of decreasing possible steric hindrance at the βCys93 reactive site. For covalent interaction to form between KAUS molecules and βCys93, the imidazole moiety of the Michael addition group has to move closer to the SH group of βCys93, while at the same time avoiding steric contact with the surrounding binding site residues. As indicated by manually docking the KAUS compounds to the binding site, such potential steric interactions would ensue unless there is a re-arrangement of the binding site residues, which could have contributed to the marginal potency of these compounds. Therefore, three subset of compounds, namely MMA-202, MMA-204, and MMA-205, without the imidazolyl group (non-imidazyl Michael addition compounds) were designed and synthesized to test this hypothesis. 

### 3.2. Chemical Synthesis

The MMA-100 series of compounds were prepared as outlined in Scheme 1, and detailed syntheses are described in the experimental section. Briefly, the compounds were synthesized by converting the methoxypicolinic acids (1a,1b) to the corresponding ketones (3a,3b) via Weinreb amide intermediate (2a,2b). The ketones were condensed with the aldehyde, 1*H*-imidazole-2-carbaldehyde, to form the Michael addition products MMA-101 and MMA-102. MMA-103 and MMA-104 were synthesized by reacting the corresponding ketones (3c,3d) with the aldehyde, 1*H*-imidazole-2-carbaldehyde under the same reaction conditions. For MMA-106 and MMA-107, the methoxy group of the ketones (3a,3b) was converted to the hydroxyl group of the ketones (4a,4b) to make the corresponding tetrahydrofuran-3-oxy group in (5a,5b), which were then reacted with the aldehyde to form the final products MMA-106 and MMA-107. The 5-acetyl-2-methoxybenzonitrile (3e) was synthesized by iodination followed by cyanation of methyl 4-methoxyphenyl ketone (6), which was condensed with the aldehyde to form MMA-105.

The MMA-200 series of compounds were prepared as outlined in Scheme 2, and detailed syntheses are described in the experimental section. MMA-202 and MMA-205 synthesis started with acylation of anisole and phenol (8a and 8b) under Lewis acid catalysis to give the ketonic intermediates (9a and 9b). The phenol (9b) was converted to 2-tetrahydropyranyl ether (10) by the reaction with 3,4-dihydro-2H-pyran (DHP) under a Brønsted acid catalysis. The 3-chloropropionyl ketones (9a and 10) were converted to the enone Michael addition final products MMA-202 and MMA-205 via treithylamine-initiated β-elimination. For the alkyne MMA-204, 4-methoxybenzoic acid (11) was coupled with N,O-dimethylhydroxylamineHCl to form the Weinreb amide (12). The amide was reacted with trimethylsilylacetylene forming (13), followed by a fluoride-induced elimination of the protective silyl group to produce the ynone final product MMA-204. The yields of the reactions described above ranged from 15% to 70%.

### 3.3. MMA Compounds Demonstrated Sustained Pharmacologic Effects In Vitro

As noted above, aromatic aldehydes suffer from oxidative metabolism by aldehyde dehydrogenase, aldehyde oxidase in the liver, blood and other tissues that leads to short half-life and suboptimal bioavailability [28,30,42,43]. In a recent study, we also showed that when incubated in whole blood in vitro, vanillin, 5-HMF, and several other aromatic aldehydes showed significant metabolism of the aldehyde [29,30,31,32,33], resulting in shortening of their allosteric and antisickling activities [32,33], which contrasts with Michael addition compounds that showed longer duration of action [36,37]. Thus, one of our main objectives for developing these compounds was to improve on the pharmacokinetic properties of aromatic aldehydes by replacing the aldehyde moiety with a more metabolically stable Michael addition reactive center, that is expected to increase the duration of action of these compounds. This was tested by incubating 2 mM of the MMA compounds with fresh normal adult whole blood (Hct of ~30%) at 37 °C as a surrogate measure of systemic metabolism, and following their effect on Hb oxygen binding affinity. At defined time points (1.5, 3, 6, 8, 12, or 24 h), aliquot samples were drawn, subsequently analyzed for their P_50_-shifts relative to the initial P_50_ value, using three-point tonometry, and the results presented in Figure 3. 

The results are shown in Figure 4 and Figure 5, and in the Table 1. The MMA-200 series of compounds showed the most potent antisickling activities (Figure 4A; Table 1). MMA-202, with the highest biological effect, inhibited sickling by 50% and 67% at 2mM and 5mM, respectively. MMA-204, the second most potent compound inhibited sickling by 42% and 63%, and lastly MMA-205 with 32% and 52% inhibitory activities, respectively. These values compare with sickling inhibition of 16% and 53% by vanillin at 2 mM and 5mM, respectively, suggesting significant improvement in the antisickling activities of the MMA-200 series of compounds especially at the lower dose of 2 mM. The slightly antisickling potency of MMA-202 over MMA-204 may suggest that the propenoyl reactive center has a slight advantage over the propynoyl reactive center since the two compounds are structurally similar. It also appears that the more bulky THP substituent of MMA-205 is not optimal for biological activity since MMA-202 and MMA-204 with smaller methyl substituent showed better antisickling activity.

The six tested MMA-100 series of compounds, including MMA-101, MMA-102, MMA-104, MMA-105, MMA-106, and MMA-107 achieved maximum P_50_ shifts of 12–18%, with MMA-102 and MMA-104 showing the most shift (Figure 3A). Nonetheless, MMA-102 and MMA-104 show significantly different activity profiles. While MMA-104 showed almost no P_50_ shift at 24 hrs from its high activity of 18% at 1.5h, the rest of the compounds appear to show sustained activity, and in fact, three of the compounds, MMA-101, MMA-102, and MMA-106 showed increasing activity even at 24 hrs (P_50_ shifts of 15%, 18%, and 15%, respectively). With the exception of MMA-105, the rest of the compounds appear to show relatively fast onset, although their activities seem to decline at the 3 h time point and then started going up again. (Figure 3A). 

Interestingly, the non-imidazyl MMA-200 compounds showed significantly more potent activity (P_50_ shifts of 24–31%) than the imidazyl MMA-100 compounds (P_50_ shifts of 12–18%), but with delayed activity with maximum P_50_ shifts at ~6h, which persisted until 8 h, followed by a decrease to the 24 hr time (Figure 3B). MMA-202 showed the highest P_50_ shift (31%), followed by MMA-204 (27%), and lastly MMA-205 (24%) (Figure 3B), indicating that the bulkier tetrahydro-2H-pyran-2-yl (THP) group had no positive impact on the overall binding and/or Hb oxygen affinity. Interestingly, propynoyl MMA-204 showed the fastest onset with a P_50_ shift of 20% at 3h, with the propenoyl MMA-202 and MMA-205 showing corresponding shifts of 17% and 11%, respectively. 

It is clear from the above study that the MMA-200 series of compounds are more potent than the MMA-100 series of compounds; however, the latter seem to show better-sustained activity in whole blood. We speculate that the MMA-100 compounds with non-terminal propenone reactive center are less susceptible to metabolism compared to the MMA-200 compounds with terminal propenoyl or propynoyl reactive centers. Nonetheless, all the MMA series of compounds appear to show better metabolic stability or longer duration effect than typical aromatic aldehydes, at least in whole blood, the latter are known to lose their activity after 2–4 hours [29,32,33]. 

Unlike the MMA-200 series of compounds, the MMA-100 series of compounds, for the most part showed significantly lower antisickling activities. Interestingly, even though MMA-102 and MMA-106 show relatively high activities, they were not dose-dependent (Figure 5A). The antisickling results show the same trend as the above oxygen affinity results, where the MMA-200 series of compounds with terminal propenoyl or propynoyl reactive centers are more biologically active than the MMA-100 series of compounds with the non-terminal 2-substituted propenone reactive center.

Using aliquots from the same incubation sample from the above antisickling study, we tested the compounds effect on Hb affinity for oxygen at 2 mM and/or 5mM, and the results are shown in Figure 4B,C, Figure 5B and Table 1. The high antisickling activity observed for the MMA-200 series of compounds are reflected in the significant increase in Hb oxygen affinity, which also follow a similar trend as the antisickling results. Figure 4C presents representative oxygen equilibrium curves of lysates from the MMA-202, MMA-204, and MMA-205 antisickling study that clearly show dose-dependent effect. At 2mM and 5mM, MMA-202 increased Hb oxygen affinity by 20% and 43%, respectively, followed by MMA-204 with 20% and 33%, and lastly MMA-250 with 11% and 29%, respectively (Figure 4B). These values compare with 12% and 28% by vanillin, respectively, suggesting significant improvement in the allosteric activity of these compounds, especially MMA-202 and MMA-204. As previously reported for allosteric effectors of Hb (T-state or R-state Hb stabilizers), the three MMA-200 series of compounds showed major effect on Hb cooperativity [30]. The shape of the OEC changed from sigmoidal to hyperbolic with increasing compound concentration (Figure 4C). At 5 mM concentration, the Hill coefficient decreased from 2.86 ± 0.7 (unliganded) to 1.96 ± 0.08 (MMA-202), 1.98 ± 0.08 (MMA-204), 2.07 ± 0.04 (MMA-205), and 2.02 ± 0.07 for the control vanillin. The reduced cooperativity is expected because of the apparent weakening of interdimer interactions in the T-state and strengthening of the interdimer interactions in the R-state with compound binding.

As expected from their relatively weak to no antisickling activities, the MMA-100 series of compounds did not show any significant effect on Hb affinity for oxygen with SS blood (Figure 5B). Clearly, the MMA-200 series of compounds are potently superior to the MMA-100 series of compounds. Likely, the MMA-200 compounds with terminal reactive center encounter less steric hindrance to the βCys93 binding site, while the MMA-100 compounds because of the non-terminal 2-substituted propenone reactive center (as part of a bridge) require significant re-arrangement of the βCys93 binding pocket to allow for effective compound interaction, resulting in reduced binding affinity for the latter compounds. 

Comparatively, the lead compounds, KAUS-33 and KAUS-38 (at 2 mM concentration) inhibited sickling by 7% and 9%, and increased Hb oxygen affinity by 15% and 12%, respectively [37]. Likewise, the aromatic aldehyde, 5-HMF (at 2 mM concentration), which recently underwent phase I/II clinical studies for the treatment of SCD but failed due to poor PK properties (as a result of metabolically unstable aldehyde) inhibited SS sickling and increase Hb oxygen affinity by 25% and 33%, respectively [28]. These observations suggest a successful outcome of our targeted structural modifications, especially with MMA-202 and MMA-204. An interesting observation worth pointing out is that despite the relatively low P_50_ shifts (20%) at 2 mM concentration, MMA-202 and MMA-204 demonstrated substantial antisickling effects of 50% and 42%, respectively. Note that this compares with 25% sickling inhibition and 33% P_50_ shift by 5-HMF, and 16% sickling inhibition and 12% P_50_ shift b.

### 3.4. MMA Compounds React and Modify Hb α-Globin and/or β-Globin

The antisickling activity of aromatic aldehydes is dependent on Schiff base interaction between the aldehyde moiety and the Hb αVal1 amines (Schiff base adduct) at the α-cleft [20,21,22,23,24,25,26,27,28,29,30,31,32,33]. while for thiols, it is dependent on disulfide bond formation between the thiol sulfur and Hb βCys93 sulfur atoms at the surface of the protein [34,35]. Michael addition compounds, like the thiols, also form covalent interaction between their β-unsaturated carbon and the sulfur atom of βCys93 to affect their antisickling activities [36,37]. Interestingly, some Michael addition compounds, e.g., KAUS-12 and KAUS-15 as mentioned above are known to also bind to αVal1 of Hb at the α-cleft in addition to their βCys93 binding, the former only observed with deoxygenated Hb, which incidentally led to stabilization of the T-state Hb to decrease the protein affinity for oxygen [36]. We investigated which Hb subunits are modified by our test compounds, by performing reverse phase HPLC analyses study on the MMA-200 series of compounds, MMA-202, MMA-204, and MMA-205 (2 mM), and the results are shown in Figure 6. The top chromatogram (untreated control sample) shows β-globin, α-globin and γ-globin peaks. The γ-globin represents ~7% of the β-like chains. The second chromatogram shows modification of both β-globin (~39%) and α-globin (~27%) by MMA-202, followed by the chromatogram for MMA-205, with ~20% modified β-globin peak, which co-elutes with the γ-globin peak (26.4% β + γ). Finally, a similar pattern is seen with MMA-204, but with only approximately 10% modified β-globin (~17% β + γ). It is clear that while MMA-204 and MMA-205 appear to bind exclusively with the β-chain, MMA-202 binds to both the α-chain and β-chain. Most likely, MMA-202 is forming covalent interactions with the αVal1 amine at the α-cleft, as well as the βCys93 sulfur. Like other Hb modifiers, binding of the MMA-200 series of compounds at the βCys93 site should lead to disruption of the T-state stabilization salt-bridge interaction between βHis146 and βAsp93, shifting the allosteric equilibrium to the R-state to increase the oxygen affinity of the protein [34,35]. The βCys93 binding site is located on the surface of the protein, and interaction with βCys93 is also expected to contribute to the antisickling activity by weakening the interactions between adjacent HbS molecules in the polymer. Of the three compounds, MMA-202 uniquely also binds to the α-globin, and most likely at the α-cleft to make covalent interaction with αVal1 amine as observed for aromatic aldehydes and the Michael addition compounds KAUS-12 and KAUS-15. However, unlike, KAUS-12 and KAUS-15 that bind to the T-state Hb and concomitantly decrease the protein affinity for oxygen, we expect MMA-202 to bind to the R-state Hb and contribute to the observed biological activities, consistent with its potent antisickling effect. 

### 3.5. MMA Compounds React with Free βCys93 of Hb

Our expectation was that the MMA compounds will bind to βCys93 to effect their antisickling activity, and although the RP-HPLC study suggests binding of the compound to the β-subunit, it does not identify βCys93 as the interacting amino acid. To determine whether βCys93 is indeed the β-subunit residue involved in the interaction with these molecules, we studied the reactivity of the MMA compounds, MMA-102, MMA-202 and MMA-204 (at 2 mM concentration) with the accessible sulfhydryl groups in Hb that was quantified by the disulfide exchange reaction of the cysteine thiols and DTNB (Ellman’s Reagent) at 412 nm. There are six cysteine residues in Hb, including a pair each of βCys93, αCys104, and βCys112, however as previously reported using mass spectroscopy, X-ray crystallography, and DTNB assay [34,35,36,37,44,45], compounds preferentially form covalent adduct with βCys93 because it is the only cysteine that is solvent exposed for reaction [36,37,44]. Consistently, the solution-based sulfhydryl assay identified only two accessible thiols, out of the six present in Hb that are likely the two βCys93 residues. As expected from their biological potency, the MMA-200 series of compounds, as exemplified by MMA-202 and MMA-204 were the most reactive, showing only about 10% freely available SH of βCys93 (i.e., forming ~90% adduct with βCys93) (Figure 7), whereas the least potent MMA-102 showed only 10% adduct formation. Ethacrynic acid (ECA), which has been studied for its antisickling activity [36,37,41], is known to form covalent adduct with βCys93 showed 70% adduct with βCys93 (Figure 7). It is clear from this study that the biological activities of the MMA compounds are in part due to their reaction with βCys93 that leads to an increasing concentration of the non-polymer forming R-state HbS. 

## 4. Conclusions

Even though aromatic aldehydes remain promising as antisickling agents, their therapeutic development into clinically useful drugs have been hampered by poor PK properties due in most part to the metabolically unstable aldehyde moiety, with the notable exception of Voxelotor. Considering the fact that Voxelotor is currently the first and only approved molecule in the class of Hb modifiers, the need remains to develop additional candidates. We have developed novel compounds with Michael addition reactive centers that not only showed superior potency over several known aromatic aldehydes, but most importantly, these compounds show improved in vitro metabolic profiles. It is expected that these properties will translate into sustained and improved pharmacologic activities in vivo.

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
