# Peer review of "An Investigation of Structure-Activity Relationships of Azolylacryloyl Derivatives Yielded Potent and Long-Acting Hemoglobin Modulators for Reversing Erythrocyte Sickling"

_biomolecules, 2020, doi:10.3390/biom10111508_

Round 1
Reviewer 1 Report
I was very pleased to read in this manuscript reports of new compounds that could lead to improvements in treatment for sickle cell disease. At this stage there remains some uncertainty about the mechanism by which the Michael addition compounds are influencing oxygen affinity and it is important to understand these mechanisms as far as possible. My concerns largely relate to the site of reaction on the Hb-alpha and Hb-beta chains. First, identification of the modified alpha/beta chains is based on the result shown in Figure 6; however, more information is needed to assign the hplc peaks to their respective chains (gel analysis, MS, etc.). Second, within each chain there are several sites that could conceivably be subject to Michael addition. Some mass spectrometry data should be obtained to provide direct evidence for the hypothesised beta-Cys93 and alpha-Val1 sites. For example, there are two additional sulfhydryls (alpha-Cys104 and beta-Cys112) where modification would be disruptive to the Hb tetramer structure. Reporting additional parameters from the oxygen saturation curves shown in Fig 4 could be informative about changes in the Hill coefficient in addition to the p50.
Author Response
The Reviewer correctly points out that there are six Hb cysteines; a pair each of βCys93, αCys104, and β112. While the solvent exposed βCys93 is known to form covalent adduct with different classes of compounds (e.g. nitrates/nitrites, thiols, Michael addition compounds, and isothiocyanates) as reported in several studies using mass spectrometry, crystallography and solution-based sulfhydryl Ellman’s (DTNB) techniques, αCys104, and β112 are buried and are not accessible to reacting with these compounds (references provided in the revision). We have performed a DTNB assay for three of the compounds, MMA102, MMA202 and MMA204, and the results show that only two accessible thiols, out of the six present in Hb, are being modified by these compounds, which are likely the βCys93 residues. The new experiment and results with appropriate citations have been included in the revised version.
The Reviewer is correct that the compounds could have an effect on Hb cooperativity as previously observed with allosteric effectors of Hb (whether increasing Hb oxygen affinity or decreasing it), due to restraining of crosstalk between the Hb subunits that is important for the R®T transition. Indeed, the increasing concentration of the compound leads to change in sigmoidal shape of the OEC to hyperbolic, and at 5 mM concentration, the Hill coefficient decreased from the normal 2.86±0.7 to 1.96 ±0.08 for MMA202, 1.98 ±0.08 for MMA204, and 2.07 ±0.04 for MMA205, and 2.02 ±0.07 for the control vanillin. The smaller Hill coefficient is expected because of the apparent weakening of interdimer interactions in the T-state and/or strengthening of the interdimer interactions in the R-state. We have accordingly revised the manuscript.
Reviewer 2 Report
In the current study, Omar et al. develop a series of compounds with anti-sickling potential. The compounds were derived by modification of lead compounds that the group has previously reported. These modifications conferred improved anti-sickling activity in in vitro assays. This study is of significant interest since there is only one other drug in this category of compounds that recently received FDA approval for treatment of sickle cell disease. While the results presented in this study are promising, further in vivo studies are required to establish pharmacokinetics and in vivo efficacy. A particular concern for potential drug candidates that increase the oxygen affinity of hemoglobin is the optimal release of oxygen to tissues, which requires in depth in vivo characterization.
Comments:
- The authors should remove “(and potentially in vivo)” from the abstract since there is no in vivo data presented to support this statement.
- Line 524: The authors suggest that these compounds may destabilize HbS polymer formation by an oxygen-independent mechanism. Can the authors experimentally demonstrate destabilization of HbS polymerization in conditions that mimic severe hypoxia?
- The label in Figure 6 should be βS-globin since SS blood was used for the assay.
- Have the authors confirmed the identity of the various globin fractions from the RP-HPLC, particularly for the modified βS-globin peaks that co-elute with γ-globin? Can this be determined by unique absorbance patterns of MMA-205 and MMA-204 (bound to Hb)?
Author Response
The authors should remove “(and potentially in vivo)” from the abstract since there is no in vivo data presented to support this statement.
Response: We agree with the Reviewer that the in vivo sentence was speculative and we have consequently removed it from the abstract. We plan to perform in vivo studies once funds and resources are available.
Line 524: The authors suggest that these compounds may destabilize HbS polymer formation by an oxygen-independent mechanism. Can the authors experimentally demonstrate destabilization of HbS polymerization in conditions that mimic severe hypoxia?
Response: This was a speculation on our part based on the fact that previously reported compounds that bind to the surface of Hb, such as to βCys93 have been suggested to prevent direct polymer destabilization, which is independent of the primary O2-dependent antisickling activity. We do not have any direct evidence to suggest that such potential polymer destabilization by these molecules could lead to O2-independent mechanism of action, and therefore we have removed the discussion of the O2-independent mechanism effect.
The label in Figure 6 should be βS-globin since SS blood was used for the assay.
Response: We have made the correction
Have the authors confirmed the identity of the various globin fractions from the RP-HPLC, particularly for the modified βS-globin peaks that co-elute with γ-globin? Can this be determined by unique absorbance patterns of MMA-205 and MMA-204 (bound to Hb)?
Response: We assigned identities to the peaks based on chain balance in hemoglobin tetramers between alpha and beta-like chains. Also, as noted above in response to a similar comment by Reviewer 1, we have performed a solution-based sulfhydryl assay (DTNB) assay that shows that the beta chain residue Cys93 is being modified.
Round 2
Reviewer 1 Report
I thank the authors for their responses and changes to the manuscript, which satisfy my previous concerns.